# Trypsin-Like Activity in Oral Cavity Is Associated with Risk of Fever Onset in Older Residents of Nursing Homes: An 8-Month Longitudinal Prospective Cohort Pilot Study

**DOI:** 10.3390/ijerph18052255

**Published:** 2021-02-25

**Authors:** Maya Izumi, Ayaka Isobe, Sumio Akifusa

**Affiliations:** School of Oral Health Sciences, Faculty of Dentistry, Kyushu Dental University, Fukuoka 803-8580, Japan; r15izumi@fa.kyu-dent.ac.jp (M.I.); r17isobe@fa.kyu-dent.ac.jp (A.I.)

**Keywords:** fever, nursing home, older adults, periodontal pathogens, trypsin-like activity

## Abstract

This study aimed to evaluate the association between trypsin-like activity in the oral cavity and the onset of fever in independent older residents of nursing homes. Independent older residents aged ≥ 65 years in 10 nursing homes were included in this study, which was conducted in Kitakyushu, Japan. For 8 months, follow-up dates on which the body temperatures of participants were more than 37.2 °C were noted. Trypsin-like activity in the oral cavity was detected by ADCHECK^®^ with five-grade evaluation at baseline. Data from 53 independent participants with median age 89.0 (67–102) years were available for analysis. ADCHECK^®^ scores were associated with fever days (r = 0.312, *p* = 0.029). The average periods until the onset of fever in participants with ADCHECK^®^ Scores 1 and 2, Score 3, and Scores 4 and 5 were 6.6 ± 0.5, 5.0 ± 0.7, and 4.1 ± 1.0 months, respectively. ADCHECK^®^ Scores 4 and 5 signified a higher risk of fever compared to ADCHECK^®^ Scores 1 and 2 (hazards ratio 5.9, 95% confidence interval 1.4–23.9, *p* = 0.013), adjusted for possible confounders. We concluded that trypsin-like activity in the oral cavity was associated with the risk of fever in independent older residents of nursing homes.

## 1. Introduction

Older adults are susceptible to infection [1]. Fever is a major indicator of infection, followed by several infectious diseases, including bacteremia, endocarditis, meningitis, nephritis, and pneumonia, and could lead to dehydration and other serious complications [2,3]. In older adults, the most common causes of fever are respiratory infections [4]. Pneumonia in older adults frequently presents without coughing or an increase in sputum production [5], and with lower-grade fever than that in young adults [6]. A previous study reported that 47% of fever episodes in older adults were of blunted fever [7]. Hence, a lower threshold of body temperature is required for monitoring the onset of fever to facilitate the early detection of infection in older adults.

According to the survey of dental diseases performed by the Ministry of Health, Labour, and Welfare in Japan in 2016, 54.5% of older adults aged ≤65 years have teeth with ≥4 mm probing pocket depths (PPDs), and the proportion of such older adults is increasing, because the average number of remaining teeth is increasing [8].

Periodontitis has been independently linked to pneumonia, cardiovascular diseases, cancer, and mortality [9,10,11,12]. Mortality due to pneumonia was 3.9 times higher in community-dwelling older adults having ≥10 teeth with ≥4-mm PPDs compared to those without periodontal pockets [12]. Periodontitis is associated with respiratory diseases due to poor oral hygiene and low immunity state [13]. A recent systematic review with a meta-analysis demonstrated that periodontitis increases the risk of pneumonia by 3.2-fold [14]. However, the relationship between onset of fever and periodontal condition or periodontal pathogens is not clearly understood. 

As fever is an indicator of infection, it is important for health professionals or caregivers to assess the risk factors for fever in older adults. Whatever the causes, the onset of fever imposes a significant burden not only on the patients, but also on medical professionals and the nursing home staff. Thus, risk factors of fever are the key to providing optimum health to geriatric individuals requiring care. Given the high prevalence of periodontitis in older adults, we hypothesized that a factor associated with trypsin-like activity in the oral cavity, putatively driven by periodontal pathogens, would be associated with the risk of fever in older residents of nursing homes, and aimed to examine trypsin-like protease activity in the oral cavity, which is considered to be associated with red-complex periodontal pathogens [15,16]. This study thus aimed to evaluate the association between trypsin-like activity in the oral cavity and the onset of fever in independent older residents of nursing homes.

## 2. Materials and Methods

### 2.1. Study Setting and Population

This prospective cohort study was performed in 10 nursing-care insurance facilities of the Kitakyushu City, Fukuoka Prefecture, Japan, from February 2019 to December 2019. The follow-up period was 8 months. Individuals who could not sit or communicate were excluded because the accurate measurement of swallowing function and tongue and lip force was required.

The study was approved by the Kyushu Dental University Institutional Review Board for Clinical Research (No. 18-51). Informed consent was obtained from all participants, their surrogates, or legal representatives before data collection.

### 2.2. Data Collection

Number of present teeth, PPD, bleeding on probing (BOP), number of functional tooth units (FTU), bacterial counts on the dorsum of tongue, swallowing function, tongue pressure, labial closing force, functional independence in activities of daily living (ADL), cognitive function, hand grip, 5 m gait, and comorbidity conditions at baseline were assessed. During the follow-up period, the dates on which a participant exhibited a body temperature of more than 37.2 °C [17], i.e., fever, were noted. Body temperature was measured daily in the axilla at 7:00 a.m. by the nursing-home staff. Demographic characteristics and physical health status were evaluated using a standard questionnaire and the medical records of the nursing-care insurance facility. All data excluding body temperature were collected from 3:00 p.m. to 5:00 p.m. at the respective nursing homes at baseline. Participants abstained from food for 2 h prior to the examinations.

### 2.3. Swallowing Function

Swallowing function was evaluated using the repetitive saliva swallowing test (RSST), which showed sensitivity and specificity of 70.0% and 71.7%, respectively, in predicting aspiration [18]. A single trained dentist (S.A.) performed the RSST. The participants were instructed to sit; the examiner used the index finger to palpate the hyoid bone and middle finger to palpate the thyroid cartilage while the participants swallowed repeatedly, and the swallowing count in 30 s was determined. A swallow was counted only when the thyroid cartilage rose past the middle finger sufficiently. If such swallows were noted three consecutive times, the swallowing function was considered normal [18].

### 2.4. Tooth and Periodontal Examination

A single trained, experienced dentist (S.A.) assessed PPDs, BOP, and number of functional tooth units (FTUs). PPDs were measured at 6 points (mesial, mid-, and distal points of buccal and lingual sites). The total number of FTUs was counted to evaluate posterior teeth occlusion. FTUs were defined as the number of pairs of natural or artificial posterior teeth in the maxillary and mandibular arches, excluding the third molars [19,20]. Scores for premolars and molars were 1 and 2 points, respectively. Accordingly, a participant with complete dentition received a perfect score of 12.

### 2.5. Maximum Tongue Pressure

Tongue pressure was evaluated using a specific tongue pressure measurement device with high reproducibility (TPM-01, JMS Co., Tokyo, Japan) [21]. All measurements were performed by a single dental hygienist (M.I.). The TPM-01 is a handheld manometry device consisting of a small, balloon-type, disposable oral probe. At zero calibration, the probe is inflated with air at pressure of 19.6 kPa. A single trained dental hygienist performed the tongue pressure measurement. Tongue pressure was measured with the participants in a relaxed and seated position. They were instructed to compress the balloon-type probe between the tongue and palate with the maximum possible force. The measurements were performed three times, and the maximum value was considered as the correct tongue pressure. The intra-examiner agreement was good (kappa values > 0.8).

### 2.6. Lip-Closure Strength

The lip-closure strength was evaluated using a digital force gauge with high reproducibility (Lipplekun, Shofu Co., Kyoto, Japan) [22]. The Lipplekun is a medical device that consists of a measuring apparatus and spindle connected to a disposable button-type intraoral probe using dental floss. The lip-closure strength in the range of 0–19.9 N was measured. A single trained dental hygienist (A.I.) performed the lip-strength measurement. Measurements were performed with the participants in a relaxed and seated position; they were instructed to grasp the button-type piece between their lips as tightly as possible. The measurements were performed three times, and the maximum value was considered to represent the correct labial closing force. The intra-examiner agreement was good (kappa values > 0.8).

### 2.7. ADL, Cognitive Activity, and Comorbidity Conditions

Trained nursing-home staff assessed the functional independence of disabled older individuals in performing activity of daily living (ADL) [23]. The criteria were not related to cognitive function. The functional independence in ADL were assessed as follows (Table 1): Rank J, almost independent living indoors and outdoors with any disabilities; Rank A, almost independent living indoors, but not outdoors; Rank B, dependent living with requirement of care, and almost bedridden in the daytime; Rank C, bedridden at all times with care required for meals, evacuation, and changing clothes. In this study, Ranks J and A were designated as “not bedridden” or “independent.”

The independence degree in daily living for the elderly with dementia was used to assess cognitive function as follows [24]: Rank I, almost independent living at home and socially, despite exhibiting cognitive impairment; Rank II, living without care but with support, despite exhibiting some interfering symptoms and behavior, and difficulty in communication; Rank III, care required, with some interfering symptoms and behavior, and difficulty in communication; Rank IV, care required at all times with frequent interfering symptoms and behaviors, and difficulty in communication; Rank M: specialized medical treatment required because of remarkable psychiatric symptoms, problematic behavior, and severe physical disorders. In this study, Ranks I and II were designated as normal, and Ranks III, IV, and M were designated as having cognitive impairment.

Comorbidity conditions were assessed using the Charlson comorbidity index [25,26], because older adults frequently live with diseases.

### 2.8. Trypsin-Like Activity and Bacterial Count

The trypsin-like activity in the oral cavity was assessed using a trypsin-activity assessment kit (ADCHECK^®^, ADTECH Co., Ltd. Oita, Japan) by inspecting the color-developing reaction [15]. ADCHECK^®^ is composed of the following three parts: a plastic board with a reaction area in a circled pit, Liquid A, and Liquid B. Red-complex periodontal pathogens (*Porphyromonas gingivalis*, *Tannerella forsythia*, and *Treponema denticola*) produce a protease, *N*-benzoyl-dl-arginine peptidase, which is responsible for the trypsin-like activity. The *N*-benzoyl-dl-arginine-2-napthylamide (BANA) substrate, incorporated in the reaction area, is enzymatically hydrolyzed by the pathogen-derived protease, to release β-naphthylamide. When one drop of chromophore liquid of ADCHECK^®^ (Liquid B) reacts with β-naphthylamide, the liquid develops a magenta color in a dose-dependent manner. The scores of ADCHECK^®^ range from 1 to 5. The scoring system was as follows: 1 was equivalent to 10 units/mL of trypsin, 2 was equivalent to 25 units/mL, 3 was equivalent to 100 units/mL, 4 was equivalent to 500 units/mL, and 5 was equivalent to 5000 units/mL. Samples were collected from the tongue dorsum of participants, using swabs, in 10 strokes. The swabs were soaked in Liquid A to suspend the sample, and then were pushed against the circled reaction area of the ADCHECK^®^ plate for 5 s. After 10 min, to allow the reaction between the BANA substrate and sample, at room temperature, one drop of Liquid B was placed on the reaction area. After an additional 3 min, the reaction area with β-naphthylamide developed magenta color in a dose-dependent manner, which was assessed using a five-grade evaluation. In this study, participants were divided into the following three groups based on the ADCHECK score: Scores 1 and 2, Score 3, and Scores 4 and 5. The trypsin-like activity was evaluated at baseline.

The bacterial count of the oral cavity was assessed using a rapid oral bacteria quantification system (Panasonic Healthcare Co. Ltd., Osaka, Japan), by performing dielectrophoresis and measuring impedance [27]. As the detection limit of this machine is 10^5^ CFU/mL, bacterial counts less than this limit were displayed as 10^5^ CFU/mL. The sampling was performed after each participant was instructed to rinse their mouth quickly with water. Oral bacteria were swabbed with three strokes of 1 cm from the tongue dorsum using a sterilized cotton swab attached to a device that facilitated even pressure. The cotton swab was inserted into measuring device [28]. These assays were performed by a single trained dental hygienist (A.I.).

### 2.9. Hand Grip and 5 m Gait

Hand grip was evaluated using a hand dynamometer (Digital dynamometer YD, Tsutsumi factory, Chiba, Japan). The grip was adjusted such that the second joint of the index finger was bent at 90 degrees. Participants were instructed to stand with their arms at their sides and grip the dynamometer without touching their bodies. The hand grip was measured twice, and the maximum value was adopted as the correct hand grip [29].

The 5 m gait was evaluated by asking the participants to walk straight ahead for 5 m after few steps [30]. A single trained dentist (S.A.) performed these tests.

### 2.10. Outcome

The primary outcome of this study was onset of fever during an 8-month period.

### 2.11. Sample Size

We did not perform the sample size estimation.

### 2.12. Bias

The cause of fever in bedridden individuals was difficult to isolate because of frailty. Hence, we recruited independent participants for the analysis.

### 2.13. Statistical Analysis

Descriptive statistics were used to characterize the participants. Each value was represented as median (minimum–maximum). The Kruskal–Wallis test was used for continuous variables, and the chi-squared test for categorical valuables. The Dann–Bonferroni test was used for post hoc analysis of the Kruskal–Wallis test. Correlation between two variables was evaluated by Spearman’s rank correlation coefficient. The incidence curves of initial fever onset during the follow-up period for participants in the three groups based on ADCHECK^®^ scores were analyzed using the Kaplan–Meier method. The Cox proportional hazards model with stepwise method was used to estimate hazard ratios (HRs) for the onset of fever among the three groups. The following variables were identified as possible confounders of the association between ADCHECK^®^ scores and the onset of fever: gender, age, cognitive function, number of teeth with BOP, and labial closing force. All analyses were performed using the IBM SPSS Statistics for Windows (IBM Corp., released 2012, version 22.0, Armonk, NY, USA), and the level of significance was set at *p* < 0.05. We followed the STrengthening the Reporting of OBservational studies in Epidemiology (STROBE) guidelines for reporting the analysis of the observational data of this study.

## 3. Results

Statistical data were obtained from 53 participants (16 males, 37 females). A total of 141 older individuals aged ≥65 years were enrolled. Ten participants refused to participate in the baseline survey, 12 participants left the nursing home before fever onset during the follow-up period, and two participants died. The records of 17 participants were incomplete. An additional 47 bedridden participants were excluded from the analysis. Finally, 53 residents were included in this study. A flow diagram illustrating the process of participant selection is provided in Figure 1. The dropout rate was 62.4%.

The median age of the participants was 88 (62–102) years. None of the participants was a current smoker. The numbers of participants with ADCHECK^®^ Scores 1 and 2, Score 3, and Scores 4 and 5 were 25 (47.2%), 21 (39.6%), and 7 (13.2%), respectively. The baseline characteristics of the three groups are shown in Table 1. The number of teeth with BOP in the group with Scores 4 and 5 was significantly higher than that in other groups (*p* = 0.007 for Kruskal–Wallis test, *p* < 0.05 for post hoc analysis comparing the Scores 4 and 5 group with Scores 1 and 2 group or Score 3 group). Similarly, labial closing force in the group with Scores 4 and 5 was significantly lower than that in the group with Scores 1 and 2 (*p* = 0.034 for Kruskal–Wallis test, *p* < 0.05 for post hoc analysis comparing Scores 4 and 5 group with Scores 1 and 2 group). Other variables indicated no significant differences among the three groups. Swallowing function was normal in all participants. No participant was hospitalized due to pneumonia during the follow-up period.

The ADCHECK^®^ scores showed moderate correlation with the number of teeth exhibiting PPDs ≥ 4 mm (r = 0.298, *p* = 0.030), number of teeth with BOP (r = 0.359, *p* = 0.008), bacterial count in the oral cavity (r = 0.290, *p* = 0.035), and fever days (r = 0.312, *p* = 0.029; Table 2).

The empirical incidence curves of initial fever during the follow-up period for the three groups are shown in Figure 2. Kaplan–Meier analysis showed that the average periods until the onset of fever in participants with ADCHECK^®^ Scores 1 and 2, Score 3, and Scores 4 and 5 were 6.6 ± 0.5, 5.0 ± 0.7, and 4.1 ± 1.0 months, respectively. A log-rank test revealed significant difference in the cumulative onset of fever between the Scores 1 and 2 and Scores 4 and 5 groups (*p* = 0.026). The incidence of initial fever in the Scores 4 and 5 group showed a stepwise increase in the initial 4 months, compared with the Scores 1 and 2 group.

In the Cox’s regression models calculated by stepwise method, the HR for initial fever in the group with ADCHECK^®^ Scores 4 and 5 was 4.2 (95% confidence interval, 1.1–15.6, *p* = 0.034) in the crude model, and 5.9 (95% confidence interval 1.4–23.9, *p* = 0.013) in the adjusted model for age, compared to the group with ADCHECK^®^ Scores 1 and 2 (Table 3).

## 4. Discussion

To the best of our knowledge, this is the first report investigating the relationship between risk of fever onset and activity of trypsin-like activity in the oral cavity in older adults. Although the causes of fever onset are unclear, considering that the most common cause of fever in patients of the rehabilitation ward was the respiratory tract [31], respiratory infection may be a major cause of fever in participants of this study as well.

We excluded frail bedridden individuals from our analysis, as opportunistic respiratory pathogen colonies are readily established in such individuals because of deterioration in salivary immune function and decreased oral commensal bacteria levels [32]. Some possible mechanisms are the suppression of oral immunity by ageing, malnutrition, or decreased physical activity, and the decrease in the numbers of oral commensal bacteria by antibiotic abuse, which facilitates the colonization of the oral cavity by respiratory pathogens [32]. Previous studies have demonstrated that *Pseudomonas aeruginosa*, an opportunistic respiratory pathogen, has the ability to interact with oral bacteria such as *Actinomyces viscosus* [33], *Streptococcus sanguinis*, *S. mitis*, and *A. naeslundii* [34], which are early colonizers of the oral biofilm, and does not have any ability to produce trypsin-like proteases. *S. agalactiae* can also interact with *A. viscosus* [32]. In these conditions, opportunistic respiratory pathogens increase in oral biofilms and can easily infect the respiratory tract and lungs [32]. Thus, these bedridden individuals were not considered suitable for an analysis meant to clarify the correlation between trypsin-like activity concerning periodontal-pathogen origin.

In the present study, 37.2 °C was adopted as a threshold for the onset of fever. Body temperature of older adults is lower than that of younger adults [1], and older adults sometimes have a blunted or delayed fever response; infected older adults may not mount a true febrile response [1]. It was reported that 37.2 °C in the morning should be regarded as the upper limit of the normal oral temperature range in healthy adults aged 40 years or younger [35]. Other studies have also adopted 37.2 °C as a threshold [36]. In older individuals, if fever is defined as a body temperature of ≥37.2°C, the sensitivity and specificity of detecting infections become 83% and 89%, respectively [17]. The Infectious Diseases Society of America recommends 37.2 °C as a threshold for onset of fever in older residents of long-term care facilities [37]. Thus, we adopted 37.2 °C as the threshold for onset of fever in this study.

In this study, trypsin-like protease activity was assessed by ADCHECK^®^, using BANA. The BANA test was developed to be used as a potential diagnostic tool for periodontal disease [38]. The sensitivity and specificity of the BANA test for BOP was 84% and 76%, respectively, in middle-aged adults [39]. In this study, the number of teeth with BOP in ADCHECK^®^ Scores 4 and 5 group was significantly higher than that in other groups. These lines of the evidence suggest that the application of an assay such as the BANA test with ADCHECK^®^ is useful to detect inflammation of periodontal tissue related to trypsin activity in the oral cavity, even in older adults.

ADCHECK^®^ was recently developed to detect red-complex pathogens by measuring the activity of *N*-benzoyl-dl-arginine peptidase (trypsin-like peptidase), using tongue-swab specimens. A tongue-swab sample would be useful to assess trypsin-like activity in whole oral cavity, because the dorsum of tongue is major reservoir of oral microorganisms. In addition, a sample from gingival fluid has a possibility to include polynuclear leucocyte-driven trypsin-like peptidase. Tongue samples can avoid the contamination of host-driven trypsin activity.

The isolation of periodontal pathogens and the detection of trypsin-like activity are not same. Our study aimed to investigate the association of trypsin-like activity to onset of fever. Since we found the association in older adults in the present study, the study of which pathogens associate with the onset of fever warrants further work.

The labial closing force in the ADCHECK^®^ Scores 4 and 5 group was significantly lower than that in Scores 1 and 2 group. Although no correlation with age is evident, labial closing force is associated with compromised hand grip, drooling of food, and increasing care needs in community-dwelling older people [40,41]. Older individuals with poor labial closing force frequently experience choking on food, suggesting that lip-closing function plays a substantial role in the pharyngeal stage of swallowing [42]. At this point, the relationship between periodontal pathogens and labial closing force is unclear; however, it can be analyzed in future studies.

Our study has several limitations. First, the causes of fever onset were not certain, because medical examination of the participants for fever was not performed. The uncertain cause of fever represents a strong limitation. The average period of fever in the participants was almost one day. The diagnosis of the cause of fever is increasingly difficult. In addition, since the pathogens associated with the onset of fever were not identified, it was unclear whether the red-complex periodontal pathogens were a direct or indirect cause of fever onset. A previous in vitro study reported that *P. gingivalis* induces inflammatory responses and promotes apoptosis in lung epithelial cells infected with the H1N1 virus [43]. Second, the number of participants, especially in the group with ADCHECK^®^ Scores 4 and 5, was limited. Future research with larger sample sizes would be required to validate the results of this study. In addition, the follow-up period was a short period of time. A longer period might be needed for robust statistical analysis. Third, cohort socioeconomic status, including years of schooling and annual income, was not evaluated. In future studies, these data should be collected and analyzed. Fourth, since the participants of this study were residents of nursing homes in one area of Japan, caution must be taken in generalizing the results. Fifth, there can also be human sources of trypsin-like activity in the oral cavity. For example, mucoid sputum includes human airway trypsin-like protease [44]. Since ADCHECK^®^ detected the human-derived trypsin-like activity, the assessment using a BANA test should be carefully interpreted. Sixth, the periodontal status was only assessed at baseline. To analyze association between onset of fever and periodontal status at the end of intervention period, the status needs to be assessed. Seventh, there may have been selection bias when the participated nursing homes were selected. In addition, as confounders, years of formal education, income, and marital history were not collected. These matters demand caution when generalizing the findings.

## 5. Conclusions

In this study, the association between trypsin-like activity in the oral cavity and risk of fever in independent older residents of nursing homes was evaluated. Older individuals with higher trypsin-like protease activity in their oral cavity had a greater number of teeth with BOP and poor labial closing force, compared to those with lower activity. Older individuals with higher ADCHECK^®^ scores had a higher risk of fever onset. The direct or indirect relationship between trypsin-like activity in the oral cavity and the onset of fever warrants further study. Further research will need to investigate the association between periodontal status and ADCHECK^®^ scores. Because ADCHECK^®^ is a rapid and simple assay kit, non-dental professionals, such as caregivers or nursing-care staff, can easily assess trypsin-like activity in dependent older adults. This assessment will contribute to the early isolation of persons with a risk of fever onset.

## Figures and Tables

**Figure 1 ijerph-18-02255-f001:**
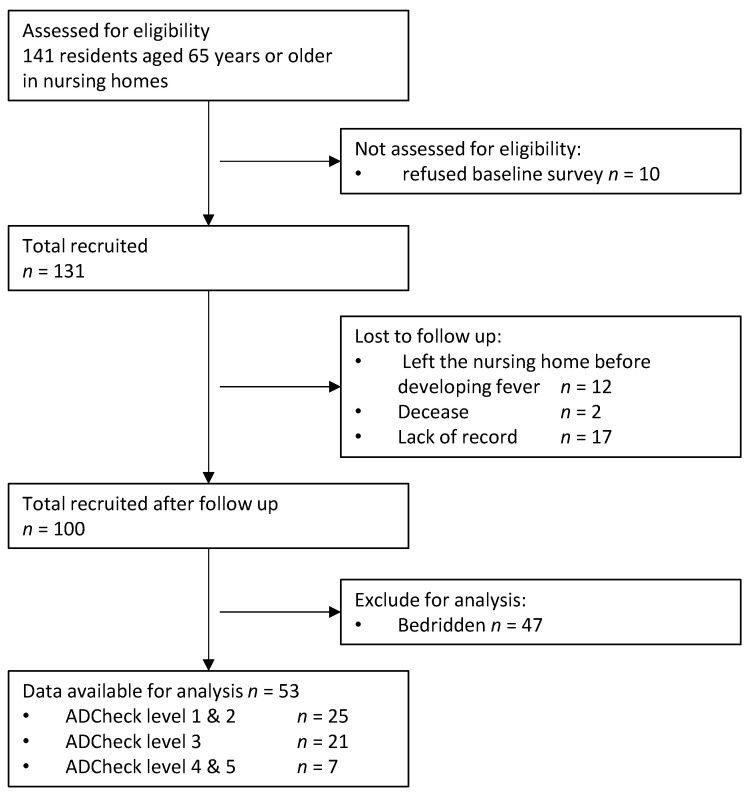
Flow diagram of the study participant selection.

**Figure 2 ijerph-18-02255-f002:**
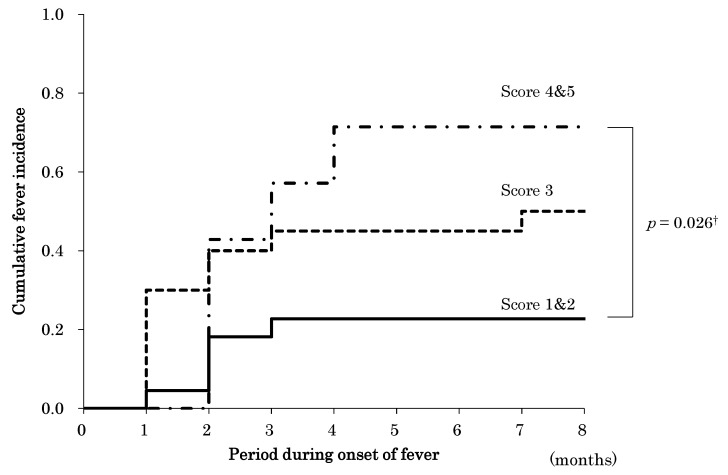
Incidence curves according to ADCHECK^®^ scores. The x-axis shows the follow-up duration (months). † = log-rank test.

**Table 1 ijerph-18-02255-t001:** Rank for activity of daily living.

Rank	Status	Classification
Rank J	Almost independent living indoors and outdoors with any disabilities	Not bedridden or independent
Rank A	Almost independent living indoors, but not outdoors	
Rank B	Dependent living with requirement of care, and almost bedridden in the daytime	Dependent
Rank C	Bedridden at all times with care required for meals, evacuation, and changing clothes	

**Table 2 ijerph-18-02255-t002:** Characteristics of each group based on ADCHECK^®^ scores.

Variables		ADCHECK^®^		*p*-Value ^§^
	Score 1 and 2 (*n* = 25)	Score 3 (*n* = 21)	Score 4 and 5 (*n* = 7)	
Age (*m*)	86 (62–94)	89 (74–102)	87 (76–93)	0.140
Gender (*n*, %) men	10 (40.0)	4 (19.0)	2 (28.6)	0.303
women	15 (60.0)	17 (81.0)	5 (71.4)	
Teeth number (*m*)	3 (0–28)	10 (0–26)	16 (5–20)	0.297
Teeth with PD > 4 (*m*)	0 (0–10)	1 (0–18)	6 (0–16)	0.077
Teeth with PD > 6 (*m*)	0 (0–3)	0 (0–3)	0 (0–12)	0.338
Teeth with BOP (*m*)	0 (0–6) *	0 (0–9) *	5 (0–10)	0.007
Functional tooth unit (*m*)	12 (0–12)	12 (0–12)	12 (0–12)	0.919
Bacterial count (*m* ×10^6^)	3.9 (0.1–20.3)	5.3 (0.7–40.1)	7 (0.1–14.6)	0.087
Lip closing force (*m*, N)	9.4 (3.6–14) *	7.6 (3–12.5)	5.5 (1–11.4)	0.034
Tongue pressure (*m*, kPa)	23 (10–38.7)	23.9 (4.5–34.9)	16 (7.3–21.3)	0.066
BMI (*m*, kg/m^2^)	20.7 (16–26.4)	22.8 (18.9–28.4)	22.2 (19.5–23.4)	0.229
Hand grip (*m*, kg)	15 (6.9–30.7)	10.6 (0–21.5)	11.8 (6.7–20)	0.062
5 m gait (*m*, s)	6.9 (3.7–24.8)	7.2 (5.5–8)	4.4 (4.2–9.1)	0.649
Cognitive function (*n*, %)	7 (28.0)	7 (33.3)	1 (14.3)	0.625
Charlson commodity index (*m*)	1 (0–4)	0 (0–2)	1 (0–1)	0.611
Fever days (*m*)	0 (0–6)	0.5 (0–11)	1 (0–4)	0.095

^§^: Kruskal–Wallis test, *: *p* < 0.05, compared to group with score 4 and 5 in ADCHECK^®^, evaluated by Dann–Bonferroni test as post hoc test. PD = pocket depth of gingiva, BOP = bleeding on probing, BMI = body mass index. m = median, n = number of persons. Each median is indicated with the minimum and maximum in parentheses.

**Table 3 ijerph-18-02255-t003:** Cox’s hazard analysis for onset of fever among groups based on ADCHECK^®^ scores.

Variables	Crude Model			Adjusted Model ^§^		
	B ± SE	HR (95% CI)	*p*-Value	B ± SE	HR (95% CI)	*p*-Value
ADCHECK^®^ scores						
Score 1 and 2	1 (reference)			1 (reference)		
Score 3	1.1 ± 0.6	3.0 (0.9–10.0)	1.1 ± 0.6	1.5 ± 0.7	4.4 (1.2–16.2)	0.025
Score 4 and 5	1.4 ± 0.7	4.2 (1.1–15.6)	0.034	1.8 ± 0.7	5.9 (1.4–23.9)	0.013
Age	−0.0 ± 0.0	1.0 (0.9–1.0)	0.500	−0.1 ± 0.0	0.9 (0.9–1.0)	0.033

Note: B = partial regression coefficient, SE = standard error, BOP = bleeding on probing, CI = confidential interval, HR = hazard ratio, **^§^** = Stepwise method.

## Data Availability

The data presented in this study are available on request from the corresponding author. The data are not publicly available due to include sensitive personal information.

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
