# Peer review of "Trypsin-Like Activity in Oral Cavity Is Associated with Risk of Fever Onset in Older Residents of Nursing Homes: An 8-Month Longitudinal Prospective Cohort Pilot Study"

_ijerph, 2021, doi:10.3390/ijerph18052255_

Round 1
Reviewer 1 Report
The paper was overall improved. However, there are some issues.
1) The authors misunderstand the sample size estimation. Based on the data in the previous references or preliminary data, the authors should calculate the sample size. However, they did not mention the reason why they estimated a rate of fever onset of 80% in one group and 40% the other groups, and its origin. Did they include dropout rate for the sample size estimation? Actually many participants dropped out in this study. The rate should be included in a cohort study. What is “13 sets”? Is the number of sample 13 participants in each group? If yes, how was the decision made to recruit 121? Why did they have to increase power even though the power set 0.80? Please add some comments in the text.
6) Please add the words, “stepwise methods” in the materials and methods, and results section (text and footnote).
Author Response
Reply to comment 1): Thank you for pointing it out. I understand what you pointed out in round 2 review. As comments in round 2, sentence “we did not perform the sample size estimation” in sample size section, and the word “a pilot study” in title were added.
Reply to comment 6): Thank you for pointing it out. The words stepwise methods” were added in statistical analysis section, results section, and footnote of TABLE 3.
Reviewer 2 Report
Dear Authors,
Thank you for addressing all the comments of the previous submission.
This work is very interesting and original. Although, what you say in the conclusion regarding the direct or indirect relationship between trypsin-like activity in the oral cavity and the onset of fever that needs further study, is true, therefore in order to easily apply this method in the future, more reasearch is needed.
Do you have maybe any suggestion to add, in order to make this possible/effective?
Author Response
Reply: Thank you for pointing it out. The sentence “More research will need to investigate the association between periodontal status and ADCHECK® scores.” was added in Conclusion section.
This manuscript is a resubmission of an earlier submission. The following is a list of the peer review reports and author responses from that submission.
Round 1
Reviewer 1 Report
Dear Authors,
I would like to thank you and congratulate you the hard work and dedication to this project. This work is very interesting and well structured.
I have few questions that I would like to have explained.
Introduction: needs to be expanded and improved. The hypothesis of correlation between the factor associated with periodontal pathogens and the risk of fever needs a wider background in order to be sustained in the affirmation of the aim and scope of the study. Furthermore, I would suggest substituting ‘we…’ with something like ‘this study/trial, aims to…’
Materials and methods:
I really appreciated how clearly materials and methods were presented.
2.1 I would suggest changing the direction of the arrows of the right-side boxes of the flow diagram, as it would better indicate the exclusion of each mentioned part.
2.3 Was the swallowing test performed by the same calibrated examiner? Please specify it.
2.5 Again, was the test performed by the same calibrated examiner? Please specify it.
2.7 Lines 121-130, my suggestion would be to arrange the different ranks in a small table, in order to make it easier and clearer to follow.
2.8 Very well explained and clear. How many examiners performed this test?
2.9 Also here, were the measurements taken by the same calibrated examiner? Please specify it.
Discussion and Conclusion
The uncertain cause of fever represents a strong limitation. Furthermore, conclusions need to be expanded and the relevance and contribution for the scientific community of the results reported should be clarified and explained.
Author Response
Reviewer 1
Dear Authors,
I would like to thank you and congratulate you the hard work and dedication to this project. This work is very interesting and well structured.
I have few questions that I would like to have explained.
Introduction: needs to be expanded and improved. The hypothesis of correlation between the factor associated with periodontal pathogens and the risk of fever needs a wider background in order to be sustained in the affirmation of the aim and scope of the study. Furthermore, I would suggest substituting ‘we…’ with something like ‘this study/trial, aims to…’
Reply: Thank you for pointing it out. Sentences regarding relationship between periodontitis and pneumonia were added. As reviewer’s valuable suggestion, “We” was changed to “This study”.
Materials and methods:
I really appreciated how clearly materials and methods were presented.
2.1 I would suggest changing the direction of the arrows of the right-side boxes of the flow diagram, as it would better indicate the exclusion of each mentioned part.
Reply: Thank you for pointing it out. The directions of the arrows were changed.
2.3 Was the swallowing test performed by the same calibrated examiner? Please specify it.
Reply: Thank you for pointing it out. Yes, a single dentist performed the swallowing test. The explanation was added.
2.5 Again, was the test performed by the same calibrated examiner? Please specify it.
Reply: Thank you for pointing it out. Yes, a single dental hygiene performed the tongue pressure measurement. The explanation was added.
2.7 Lines 121-130, my suggestion would be to arrange the different ranks in a small table, in order to make it easier and clearer to follow.
Reply: Thank you for pointing it out. Table for explanation of ranks of ADL was added as TABLE 1.
2.8 Very well explained and clear. How many examiners performed this test?
Reply: A single examiner performed each test. The number of researchers collected data was 3.
2.9 Also here, were the measurements taken by the same calibrated examiner? Please specify it.
Thank you for pointing it out. Yes, a single examiner performed each test.
Discussion and Conclusion
The uncertain cause of fever represents a strong limitation. Furthermore, conclusions need to be expanded and the relevance and contribution for the scientific community of the results reported should be clarified and explained.
Thank you for pointing it out. As reviewer pointed out, the uncertain cause of fever represents a limitation, although the outcome of this study is onset of fever regardless underlying disease. The sentence “The uncertain cause of fever represents a strong limitation” was added.
As reviewer’s valuable comments, conclusion section was expanded.
Reviewer 2 Report
The Authors must see my remarks
Some References are missing
The title must be changed as the Authors investigated PD indices.....

Author Response
Thank you for your comments.
Reply to comments were described in annotations which were added by reviewer 2 in pdf file.

Reviewer 3 Report
The paper shows that among Japanese elderly of nursing homes, trypsin-like activity on tongue was associated with fever onset in a short-term cohort study.
This is an interesting study. However, I would like to make some points regarding the manuscript. The article needs to be revised. First, the authors should follow the STROBE guideline totally. Second, the logic is unclear. Third, the dropout rate is high. Fourth, there are no data of periodontal pathogens.
INTRODUCTION
1) One paragraph should include one topic.
2) The logic is unclear. Why did the authors focus on elderly of nursing homes, trypsin-like activity on tongue, and fever? Why not respiratory infections or Pneumonia? Furthermore, they did not investigate periodontal pathogens. Trypsin-like activity on tongue is not refer them directly and not specific. Why did they include oral function, FIU, and Hand grip and 5-m gait?
3) Please refer the recent papers including doi: 10.3390/dj8030098.
MATERIALS AND METHODS
1) The authors should follow the STROBE guideline. Some important factors of STROBE checklist are missed. Please add more in the text; inclusion/exclusion criteria, bias, sample size estimation, diagnostic criteria, missing data, etc.
2) Please move the Figure 1 and related comments to result section. Please refer the STROBE checklist.
3) Why the authors select 37.2 as a cut-off point? Please add some explanations and references in the text.
4) Who did perform each examination and is a dentist assessed periodontal condition? Please add details in the text.
5) Please add the comments about validity, reliability, reproducibility and error in each measurement.
6) Please add the timing for each examination. It would be at baseline only, except for fever onset. Furthermore, the authors should add exact time of sampling and any conditions before sampling.
7) The authors should perform Full Information Maximum Likelihood method or Multiple Imputation method because dropout rate is high.
8) The outcome should be onset of fever but not “onset of fever correlated with trypsin-like activity in the oral cavity”.
9) Please changed the confounders based on the references and these should be selected when the p value was < 0.20 for the Mann–Whitney U test, chi-squared test, or Fisher’s exact test in each variable and based on previous studies because potential confounders should be eliminated only if p > 0.20 in order to prevent residual confounding. Furthermore, valuables are too much in the model. Thus, 95%CI is quite wide and the model is not appropriate. Please consider stepwise methods.
RESULTS
1) Please revise the results based on new methods.
DISCUSSION
1) Please revise the discussion based on new results.
2) Please add other comments about limitations, such as no data of changes in important factors during the period, and no data of trypsin-like activity in GCF or saliva, and no data of periodontal pathogens.
Author Response
Reviewer 3
The paper shows that among Japanese elderly of nursing homes, trypsin-like activity on tongue was associated with fever onset in a short-term cohort study.
This is an interesting study. However, I would like to make some points regarding the manuscript. The article needs to be revised. First, the authors should follow the STROBE guideline totally. Second, the logic is unclear. Third, the dropout rate is high. Fourth, there are no data of periodontal pathogens.
INTRODUCTION
1) One paragraph should include one topic.
Thank you for pointing it out. As reviewer pointed out, paragraphs were changed.
2) The logic is unclear. Why did the authors focus on elderly of nursing homes, trypsin-like activity on tongue, and fever? Why not respiratory infections or Pneumonia? Furthermore, they did not investigate periodontal pathogens. Trypsin-like activity on tongue is not refer them directly and not specific. Why did they include oral function, FIU, and Hand grip and 5-m gait?
Thank you for pointing it out. Whatever the causes, the onset of fever imposes a significant burden not only on the patients but also on medical professionals and the nursing home staff. Thus, risk factors of fever are the key to provide optimum health to geriatric individuals requiring care. Given that high prevalence of periodontitis in older adults, we hypothesized that trypsin-like activity in oral cavity, putative driven from periodontal pathogens, had direct or indirect effect of on onset of fever in older residents of nursing home. The words “periodontal pathoge” was changed to “trypsin-like activity in oral cavity, putative driven from periodontal pathogens” in revised manuscript.
ADCHECK is recently developed to detect red-complex pathogens, i.e., Porphyromonsa gingivalis, Treponema denticola, and Tannerella forsythia by measuring the activity of N-benzoyl-dl-arginine peptidase (trypsin-like peptidase), using tongue swab specimens. Tongue swab sample would be useful to assess trypsin-like activity in whole oral cavity, because the dorsum of tongue is major reservoir of oral microorganisms. In addition, sample from gingival fluid has a possibility to include polynuclear leucocyte-driven trypsin-like peptidase. Tongue sample can avoid the contamination of host-driven trypsin activity.
As reviewer pointed out, the isolation of periodontal pathogens and the detection of trypsin-like activity is not same. The study aimed to investigate the association of trypsin-like activity to onset of fever. Since we found the association in older adults in the present study, the study on which pathogens associate to onset of fever warrants further work.
Oral function including functional tooth units, hand grip, and 5-m gait speed are indicators of frailty. These are putative confounding factor between trypsin-like activity and onset of fever in older adults.
3) Please refer the recent papers including doi: 10.3390/dj8030098.
Reply: As reviewer’s variable suggestion, the reference was added as [15].
MATERIALS AND METHODS
1) The authors should follow the STROBE guideline. Some important factors of STROBE checklist are missed. Please add more in the text; inclusion/exclusion criteria, bias, sample size estimation, diagnostic criteria, missing data, etc.
Reply: Thank you for pointing it out. Inclusion/exclusion criteria, bias, sample size estimation were added. Diagnostic criterion for fever onset was 37.2°C, described in 2.2 Data collection section.
2) Please move the Figure 1 and related comments to result section. Please refer the STROBE checklist.
Reply: Thank you for pointing it out. Figure 1 and related comments were moved to results section.
3) Why the authors select 37.2 as a cut-off point? Please add some explanations and references in the text.
Reply: Thank you for pointing it out. The reference was described in 2.2. data collection section, and the cut-off point was discussed in Discussion section.
4) Who did perform each examination and is a dentist assessed periodontal condition? Please add details in the text.
Reply: Thank you for pointing it out. Examiner for periodontal condition was described in 2.4. Tooth and periodontal examination.
5) Please add the comments about validity, reliability, reproducibility and error in each measurement.
Reply: Thank you for pointing it out. Based on the references and specification sheet of devices for tongue pressure measurement and lip-close force, comments for reproducibility were added.
6) Please add the timing for each examination. It would be at baseline only, except for fever onset. Furthermore, the authors should add exact time of sampling and any conditions before sampling.
Reply: Thank you for pointing it out. The timing for examinations and condition of participants were described in 2.2. Data collection section.
7) The authors should perform Full Information Maximum Likelihood method or Multiple Imputation method because dropout rate is high.
Reply: Thank you for pointing it out. Although we recognize the utility of imputation method, these methods occasionally increase bias. Thus, in the study, we did not perform these statistical methods.
8) The outcome should be onset of fever but not “onset of fever correlated with trypsin-like activity in the oral cavity”.
Reply: Thank you for pointing it out. As reviewer’s comment, the sentence was changed.
9) Please changed the confounders based on the references and these should be selected when the p value was < 0.20 for the Mann–Whitney U test, chi-squared test, or Fisher’s exact test in each variable and based on previous studies because potential confounders should be eliminated only if p > 0.20 in order to prevent residual confounding. Furthermore, valuables are too much in the model. Thus, 95%CI is quite wide and the model is not appropriate. Please consider stepwise methods.
Reply: Thank you for pointing it out. As reviewer’s comment, the Cox hazard analysis was recalculated, shown in revised Table 4.
RESULTS
1) Please revise the results based on new methods.
Reply: Thank you for pointing it out. As reviewer’s comment, the Cox hazard analysis was recalculated, shown in revised Table 4.
DISCUSSION
1) Please revise the discussion based on new results.
Reply: Based on the new results, there were no change in the discussion.
2) Please add other comments about limitations, such as no data of changes in important factors during the period, and no data of trypsin-like activity in GCF or saliva, and no data of periodontal pathogens.
Reply: Thank you for your variable comments. Comment on no data of changes in periodontal status during the period was added as sixth limitation. Comment on no data of periodontal pathogens was described in first limitation. GCF has a possibility of trypsin-like peptidase driven from polynuclear leucocyte.

Round 2
Reviewer 1 Report
Dear Authors,
Thank you for the dedication you put in reviewing your work and addressing my comments. The paper is overall improved and clearer, however, before publishing it, minor adjustments can be made.
1. Lines 61-63 and 65-67 there is a repetition of the same period, please delete the double.
2. Line 79, I believe is better writing 'hour' as extended word instead of 'h'
3. Please add the initials of each examiner when mentioned in the measurements he/she was in charge of, and maybe numerate the examiners (both dentists and denta hygienists).
4. Why did you choose the Schoenfeld test? Why didn't you choose for example the chi-squared test?
5. Careful! The arrows of the diagram in Figure 1, weren't actually changed in the image in the text. The arrows of the squares of subjects excluded in each phase should be pointing towards right, so towards the outside of the diagram.
Reviewer 3 Report
he paper was overall improved. However, there are some issues.
1) The authors answered as below.
“Thank you for pointing it out. Whatever the causes, the onset of fever imposes a significant burden not only on the patients but also on medical professionals and the nursing home staff. Thus, risk factors of fever are the key to provide optimum health to geriatric individuals requiring care. Given that high prevalence of periodontitis in older adults, we hypothesized that trypsin-like activity in oral cavity, putative driven from periodontal pathogens, had direct or indirect effect of on onset of fever in older residents of nursing home. The words “periodontal pathoge” was changed to “trypsin-like activity in oral cavity, putative driven from periodontal pathogens” in revised manuscript.
ADCHECK is recently developed to detect red-complex pathogens, i.e., Porphyromonsa gingivalis, Treponema denticola, and Tannerella forsythia by measuring the activity of N-benzoyl-dl-arginine peptidase (trypsin-like peptidase), using tongue swab specimens. Tongue swab sample would be useful to assess trypsin-like activity in whole oral cavity, because the dorsum of tongue is major reservoir of oral microorganisms. In addition, sample from gingival fluid has a possibility to include polynuclear leucocyte-driven trypsin-like peptidase. Tongue sample can avoid the contamination of host-driven trypsin activity.
As reviewer pointed out, the isolation of periodontal pathogens and the detection of trypsin-like activity is not same. The study aimed to investigate the association of trypsin-like activity to onset of fever. Since we found the association in older adults in the present study, the study on which pathogens associate to onset of fever warrants further work.
Oral function including functional tooth units, hand grip, and 5-m gait speed are indicators of frailty. These are putative confounding factor between trypsin-like activity and onset of fever in older adults.”
Please add major points in the text.
2) There are many types of bias. Which bias is treated in this study? Please add some more comments in the text. Then, if there are others, please add the comments in the limitation.
3) Please add the name of examiner in each measurement.
4) Please add the comments about validity, reliability, reproducibility and error in each measurement (reminder). Please add some data; i.e., intra-examiner agreement was good (kappa values >0.8) and/or the reproducibility was confirmed and the error was within 5%, etc.
5) What is Schoenfeld test? It is not used in this study. Please use chi-squared test or Dann-Bonferroni test. Furthermore, nobody confirms the sample size estimation in the current form. Please add more, such as references, outcome, dropout rate. If there are no appropriate references, the authors should add “a pilot study” and “we did not perform the sample size estimation” in the title and text.
6) The valuables are too much in the model. Thus, 95%CI is quite wide and the model is not appropriate. Please consider stepwise methods (reminder).